# Natural Gas Conversion and Organic Waste Gasification by Detonation-Born Ultra-Superheated Steam: Effect of Reactor Volume

**Sergey M. Frolov \*, Viktor A. Smetanyuk, Ilias A. Sadykov, Anton S. Silantiev, Igor O. Shamshin, Viktor S. Aksenov, Konstantin A. Avdeev and Fedor S. Frolov**

Department of Combustion and Explosion, Semenov Federal Research Center for Chemical Physics of the Russian Academy of Sciences, 119991 Moscow, Russia; smetanuk@mail.ru (V.A.S.); ilsadykov@mail.ru (I.A.S.); silantevu@mail.ru (A.S.S.); igor_shamshin@mail.ru (I.O.S.); v.aksenov@mail.ru (V.S.A.); kaavdeev@mail.ru (K.A.A.); f.frolov@chph.ru (F.S.F.)
\* Correspondence: smfrol@chph.ras.ru

**Abstract:** The pulsed detonation (PD) gun technology was applied for the *autothermal* high-temperature conversion of natural gas and atmospheric-pressure oxygen-free *allothermal* gasification of liquid/solid organic wastes by detonation-born ultra-superheated steam (USS) using two flow reactors of essentially different volume: 100 and 40 dm$^3$. Liquid and solid wastes were waste machine oil and wood sawdust, with moisture ranging from 10 to 30%wt. It was expected that decrease in the reactor volume from 100 to 40 dm$^3$, other conditions being equal, on the one hand, should not affect natural gas conversion but, on the other hand, could lead to an increase in the gasification temperature in the flow reactor and, correspondingly, to an increase in the product syngas (H$_2$ + CO) quality. The PD gun was fed by natural gas–oxygen mixture and operated at a frequency of 1 Hz. As was expected, complete conversion of natural gas to product syngas in the PD gun was obtained with H$_2$/CO and CO$_2$/CO ratios equal to 1.25 and 0.25, irrespective of the reactor volume. Liquid and solid wastes were gasified to H$_2$, CO, and CH$_4$ in the flow reactors. The steady-state H$_2$/CO and CO$_2$/CO ratios in the syngas produced from waste machine oil were 0.8 and 0.5 for the 100-dm$^3$ reactor and 0.9 and 0.2 for the 40-dm$^3$ reactor, respectively, thus indicating the expected improvement in syngas quality. Moreover, the maximum mass flow rate of feedstock in the 40-dm$^3$ reactor was increased by a factor of over 4 as compared to the 100-dm$^3$ reactor. The steady-state H$_2$/CO and CO$_2$/CO ratios in the syngas produced from the fixed weight (2 kg) batch of wood sawdust were 0.5 and 0.8 for both reactors, and the gasification time in both reactors was about 5–7 min. The measured H$_2$ vs. CO$_2$ and CO vs. CO$_2$ dependences for the syngas produced by the *autothermal* high-temperature conversion of natural gas and atmospheric-pressure *allothermal* gasification of liquid/solid organic wastes by USS at *f* = 1 Hz were shown to be almost independent of the feedstock and reactor volume due to high values of local instantaneous gasification temperature.

**Keywords:** conversion of natural gas; gasification of liquid/solid wastes; steam; carbon dioxide; technology of pulsed detonation gun



## 1. Introduction

Basic technologies for the noncatalytic thermal conversion of natural gas and gasification of organic wastes are presented, e.g., in [1] and [2,3], respectively. The most valuable product of conversion/gasification is syngas. In general, syngas contains H$_2$, CO, CO$_2$, and non-condensable hydrocarbons like CH$_4$, C$_2$H$_6$, etc. On average, a low heating value (LHV) of syngas is 15–17 MJ/kg (12–16 MJ/nm$^3$). The syngas quality is higher if it contains more H$_2$ and CO and less CO$_2$. The amount of H$_2$, CO, and CO$_2$ in syngas depends on the gasifying agent, feedstock molecular structure, gasification temperature, and many other factors, and the H$_2$/CO and CO$_2$/CO ratios can vary over a wide range. Depending on the

H$_2$/CO ratio in syngas, it can be suitable for various applications. Thus, syngas with large H$_2$/CO ratios can be used for NH$_3$ synthesis or for H$_2$ production. Syngas with H$_2$/CO = 1–2 is good for production of methanol and motor fuels. As for the CO$_2$/CO ratio, it is treated as a measure of syngas contamination and must be maintained as low as possible. Syngas production by waste gasification can be accompanied by the formation of tar and char, which can contain a significant amount of carbon. The gasification process is treated as more efficient if carbon conversion is higher, i.e., if the yield of syngas is higher at less by-products formed.

The gasification processes of liquid/solid wastes in the H$_2$O/CO$_2$ environment, i.e., in the absence of free oxygen, are known to include endothermic and exothermic heterogeneous reactions [2–4]:

$$C + H_2O = CO + H_2$$
$$C + CO_2 = 2CO$$
$$C + 2H_2 = CH_4$$
$$2C + 2H_2O = CH_4 + CO_2$$

$\qquad$ (1)

and gas-phase reactions:

$$CO + H_2O = CO_2 + H_2$$
$$CO + 3H_2 = CH_4 + H_2O$$
$$CO_2 + 4H_2 = CH_4 + 2H_2O$$
$$2CO + 2H_2 = CH_4 + CO_2$$
$$C_nH_m + nH_2O = nCO + (n + 0.5\ m)\ H_2$$
$$C_nH_m + nCO_2 = 2nCO + 0.5\ mH_2$$

$\qquad$ (2)

The rates of these reactions are highly sensitive to the gasification temperature, while the degree of their completion depends on the gas and condensed-phase residence times in the flow reactors.

All known technologies for noncatalytic thermal conversion of natural gas and gasification of organic wastes can be classified into two groups, namely, low-temperature (800–1300 K) and high-temperature (above 1500 K) [4]. The former technologies are characterized by low-quality syngas [5,7,9–11], complicated control of gas quality caused by long feedstock residence times in the reactor [5,8,11,12], low overall efficiency due to high tar and char contents, and low syngas yields [5,7–15]. As for the latter technologies, they provide a syngas of high-quality [16–20], high carbon conversion efficiencies due to small tar and char residues [17,20], easy control of gas quality due to short feedstock residence times in the reactor [16–18,21], and high overall efficiency [17,20]. Electric arc and microwave plasma gasification technologies, as well as combustion- and detonation-based gasification technologies, are considered the most promising high-temperature technologies. Such technologies allow processing of wastes of arbitrary morphological and chemical composition and provide their full utilization without harmful emissions into water bodies and atmosphere. The temperature of feedstock gasification in plasma reactors is usually below 2000 K [19,22,23], indicating that the energy-consuming gas–plasma transition is an unnecessary intermediate stage. Moreover, plasma technologies require expensive structural materials for gasifier walls. Technologies of feedstock gasification by ultra-superheated steam (USS) with a temperature on a level of 2000 K obtained by burning environmentally clean H$_2$–O$_2$ mixture [24], or a mixture of fuel gas with oxygen-enriched (up to 60%vol) air [25], or a syngas–steam–oxygen mixture [26], are competitive to the plasma technologies as they do not involve energy losses inherent in gas–plasma transition. Unfortunately, these technologies require sustained thermal insulation of combustors and gasifiers and have not yet been implemented. Gasification temperatures provided by detonation-based technologies are also on a level of 2000 K and above, but do not require the energy-consuming gas–plasma transition and special structural materials. The first mention of detonation technology goes back to [27,28]. A single-pulse detonation-induced flow was used to gasify pulverized coal particles 10 to

1000 µm in diameter in the hot detonation products for a time of less than 15 ms. A carbon conversion efficiency of 40% was attained in several experiments. A pulse detonation device for *autothermal* coal gasification was patented in [29]. The device aimed to discharge high pressure oxygen and detonation-born steam into a gasifier with coal bed to avoid coal bed slagging, support coal combustion, and enhance the gasification process of the remaining coal carbon in gasification reactions with carbon dioxide, steam, and hydrogen. The important benefit of detonation over conventional combustion is that in addition to higher thermal energy, it provides the reaction products with a considerable kinetic energy. This kinetic energy can be utilized for enhancing feedstock gasification by means of shock-induced feedstock fragmentation and efficient interphase mixing, heat, and mass transfer. The ratio of the kinetic energy of the detonation products to their enthalpy at the Chapman–Jouguet plane in the fixed set of coordinates is approximately $(\gamma - 1)/2\gamma^2$ [30], where $\gamma$ is the ratio of specific heats. At $\gamma \approx 1.3$, the kinetic energy share in the propagating detonation wave is about 9%.

The objective of this work is to demonstrate the technology of a pulsed detonation (PD) gun for *autothermal* high-temperature conversion of natural gas and oxygen-free *allothermal* gasification of organic wastes with detonation-born USS using two flow reactors of essentially different volume (100 and 40 dm$^3$) at the same operation frequency of the PD gun (1 Hz). Currently, 1 Hz is the maximum frequency of the available PD gun operating on natural gas–oxygen–steam mixture. This frequency is limited by the oxygen flow rate provided by the existing liquid oxygen cryocylinder. It is implied that the PD gun technology will allow the maintenance of a highly reactive environment in a gasifier due to very high operation temperature, intense detonation-induced crunching and mixing of feedstock, and high-speed convective flows. These distinctive and unique features of the technology can potentially ensure the complete conversion of gas, liquid, and solid wastes into useful products—syngas, consisting exclusively of hydrogen and carbon monoxide, fine particles of mineral residues, consisting of safe simple oxides, as well as aqueous solutions of simple acids such as HCl, HF, $H_2S$, etc., and ammonia $NH_3$. Importantly, a part of product syngas (ideally about 10% [31]) can be used for replacing natural gas (starting fuel) for USS production in the PD gun, whereas the rest can be used for downstream needs.

The technology of PD gun was first proposed in [32–34] and substantiated in [31,35–38]. At a given operation frequency of the PD gun, both the local instantaneous temperature and time averaged mean temperature of USS in the 40-dm$^3$ reactor are expected to be higher than in the 100-dm$^3$ reactor, which must manifest itself in the improvement of product syngas quality and process efficiency.

## 2. Methods and Materials

### 2.1. Technology

The PD gun technology is described in detail in [4,31]. The gasifier contains two main units, the PD gun and flow reactor, and is equipped with systems of fuel, oxygen, and steam supply, as well as a feedstock feeder and syngas dryer. The PD gun is generally fed by a pressurized mixture of syngas, oxygen, and steam. To bring the gasifier to the operation mode, a starting fuel (e.g., natural gas) is used, which is then replaced by the syngas. Feedstock can be supplied to the reactor either separately or with supersonic jets of detonation products and is gasified in intense vortical flows inside the reactor. The syngas can be partly used for feeding the PD gun, while the remaining part can be used for external needs (production of chemicals, electricity, heat.).

Figure 1 shows schematics of two experimental gasifiers used in this study. Each gasifier consists of two communicating units: PD gun in the form of a round tube with tube branching (Y-section), and a spherical (Figure 1a) or cylindrical (Figure 1b) flow reactor. The PD gun periodically (with a frequency of $f \geq 1$ Hz) generates and transmits strong shock waves and supersonic jets of detonation products to the flow reactor. Since the PD gun operates on deflagration-to-detonation transition, it utilizes very low energy for detonation

initiation. The detonation products of natural gas–oxygen–steam mixture represent a blend of up to 80%vol $H_2O$ and up to 20%vol $CO_2$ with a temperature above 2200 K [4].

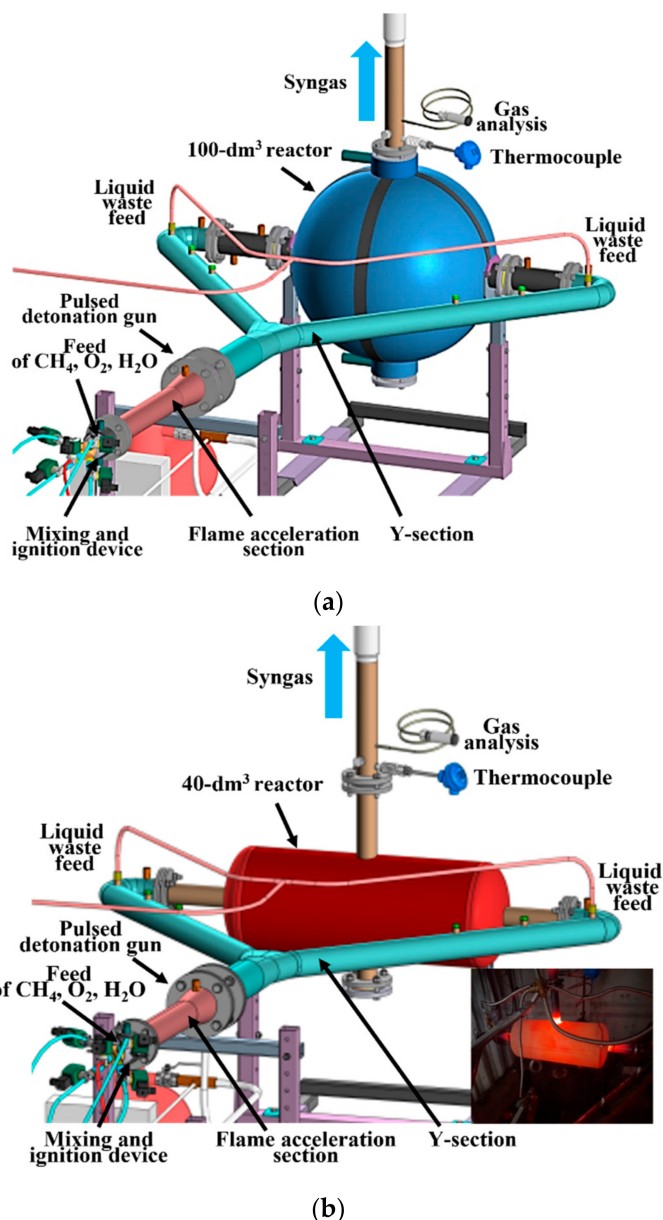

**Figure 1.** Schematics of two organic waste gasifiers with a PD gun and (**a**) spherical 100-dm$^3$ reactor and (**b**) cylindrical 40-dm$^3$ reactor.

Shock waves and jets of the detonation products form intense vortices in the flow reactor and facilitate gasification of particles of crushed waste. The syngas continuously flows out of the reactor through the outlet tube. An overpressure of 0.01–0.03 MPa was maintained in the flow reactor to avoid suction of atmospheric air. To maintain the wall temperature $T_w$ at a level $373 \leq T_w \leq 420$ K, the PD gun was equipped with a water-cooling jacket. The spherical reactor of Figure 1a was also equipped with a water-cooling jacket. As for the cylindrical flow reactor of Figure 1b, it was purposefully not water cooled and could warm up to a red glow (see insert in Figure 1b) for increasing the operation temperature. Thus, at a given operation frequency of the PD gun, both the local instantaneous temperature and time averaged mean temperature of USS in the 40-dm$^3$ reactor must be higher than in the 100-dm$^3$ reactor for two reasons: (i) due to a lower volume, and (ii) due to the absence of water cooling. The higher temperatures in the

40-dm$^3$ reactor must manifest itself in the improvement of product syngas quality and process efficiency.

Multidimensional calculations of flow structure in the reactor [31,36] indicate that the mean temperature in the spherical flow reactor exhibits a sawtooth-like time history with the time-averaged values of about 1200 and 2200 K when the PD gun operates with $f$ = 1 and 5 Hz, respectively. The sawtooth-like time history of the mean USS temperature in the reactor is caused by the periodic release of shock waves and supersonic USS jets followed by USS expansion to a pressure of $P$ = 0.1 MPa and cooling on the reactor walls. Additionally, calculations show that the time-averaged mean USS temperature in the flow reactor is not much affected by the reactor wall temperature, implying that the walls of the flow reactor can be water cooled and manufactured from conventional structural materials.

### 2.2. Gasifier

All systems of the gasifier are described in detail in [31]. Briefly, the PD gun is a tube of 50 mm i.d. with walls 2.5 mm thick. It includes three sections: a mixing/ignition section, a flame acceleration section 0.9 m long, and the Y-section containing two branches in the form of tubes 50 mm i.d. and 1.4 m long (see Figure 1). The mixing/ignition section is a mixing head equipped with feed manifolds for oxygen, natural gas, and steam, and two spark plugs (ignition energy ~100 mJ). Oxygen and natural gas are fed from the pressurized oxygen and fuel receivers connected with liquid oxygen cryocylinder (volume 495 dm$^3$) and natural gas cylinders (volume 600 dm$^3$), respectively. Steam is fed from the electrical steam generator via a heat-insulated line. Steam is used for separating hot detonation products and fresh fuel mixture to prevent parasitic ignition. The flame acceleration section ensures fast transition from deflagration to detonation in natural gas–oxygen mixture and the formation of a steady detonation wave. The flame velocity increases by three orders of magnitude in this section: from 2–3 m/s to ~2.4 km/s. A steam supply manifold is installed at the end of this section to form a natural gas–oxygen–steam mixture in the attached Y-section.

The Y-section splits the arising detonation wave into two detonation waves. These waves propagate along the branch tubes in natural gas–oxygen–steam mixture and enter synchronously the flow reactor. Spray injectors for continuous feed of liquid waste are installed at the end of each branch tube. The injectors are connected to a nitrogen-pressurized 6-dm$^3$ tank by manifolds, providing a liquid mass flow rate of up to 14 g/s. At the end of each branch tube, a provision is made for installing screw feeders of crushed solid waste. Liquid and solid wastes undergo fine aerodynamic fragmentation in periodic detonation waves and enter the reactor with jets of hot detonation products. A batch of solid waste can be also loaded directly into the flow reactor prior to gasification.

The spherical and cylindrical flow reactors are made of conventional steel and can withstand an internal explosion with an overpressure of ~1.4 MPa. Both flow reactors are equipped with four flanges. The lower flange is used for visual inspection, loading batches of solid waste, and removing solid mineral residue. Two opposite horizontal flanges are connected to the branch tubes of the Y-section. The upper flange is connected to the outlet tube for syngas removal via a separator device preventing waste particles from leaving the reactor with syngas. To demonstrate the temperature patterns and the way of its change in the reactor, Figure 2 shows the calculated snapshots of gas temperature during a time interval of 0.1 s after one of multiple detonation shots in the 100-dm$^3$ spherical reactor partly filled with 1-mm-size waste particles. The calculation is made using the approach discussed in [31,36]. As seen, the local instantaneous temperatures inside the reactor change very rapidly and attain values between 1500 and 2500 K. Since the measurements of rapidly changing local instantaneous temperatures inside the reactor are very problematic, the mean syngas temperature is measured downstream of the separator device by the thermocouple installed in the outlet tube.

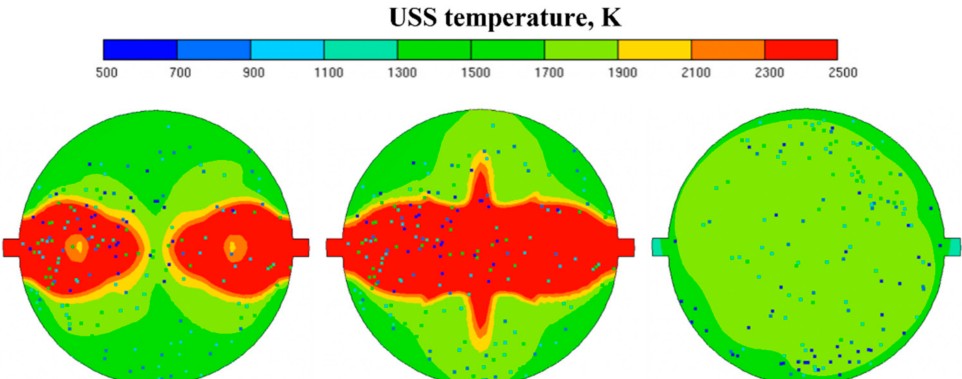

**Figure 2.** The calculated snapshots of gas temperature during a time interval of 0.1 s after one of multiple detonation shots in the 100-dm$^3$ spherical reactor partly filled with 1-mm-size waste particles. Dots represent waste particles; dot color represents particle residence time; $f$ = 5 Hz, $P \approx 0.1$ MPa.

The product syngas was first dried and cooled to 40 °C in a tubular cooler, and then entered the cyclone, where condensed and gas phases are separated. Thereafter, a part of cooled dry syngas goes to the sample preparation line of the gas analysis system, while another part is directed to a diffusion burner as a temporary utilization solution. According to estimates [31], about 10% (ideally) of the syngas is needed for replacing natural gas for USS production in the PD gun, whereas the rest can be used for downstream needs. After purification in the sample preparation line, the syngas enters the sensors of the flow gas analyzer MRU VARIO SYNGAS PLUS (Germany), recording volume fractions of $H_2$, CO, $CO_2$, $CH_4$, and $O_2$. The measurement error is estimated at 5%vol.

Gasifier operation includes the starting and steady-state modes. In the starting mode, the gasifier thermal state and the composition of cooled dry detonation products are monitored, while steam and waste are not fed. When water temperature in the cooling jackets reaches 110–140 °C and the composition of detonation products stops changing with time, the starting mode switches to a steady-state mode. The time taken for the gasifier to attain the steady-state mode is ~5 min. After the steady-state mode is reached, steam and waste are fed to the gasifier, and the gas analyzer starts monitoring the syngas composition.

*2.3. Materials*

In the first of three series of experiments, the *autothermal* high-temperature conversion of natural gas was studied. For this purpose, the PD gun was operated on natural gas–oxygen mixtures of different fuel-to-oxygen equivalence ratio, $\Phi$, and the corresponding chemical compositions of cooled dry detonation products were determined. The natural gas composition is shown in Table 1.

**Table 1.** Composition of natural gas.

| Species | %vol |
|---|---|
| $CH_4$ | 96.1 |
| $C_2H_6$ | 2.1 |
| $C_3H_8$ | 0.6 |
| $C_4H_{10}$ | 0.2 |
| $N_2$ | 1.0 |

In the second and third series, experiments on atmospheric-pressure oxygen-free *allothermal* gasification of liquid/solid organic wastes by detonation-born USS were performed. Liquid and solid wastes were represented by waste machine oil and wood (birch) sawdust, respectively. Table 2 shows the proximate and ultimate analyses of the wastes. The elemental analysis was made with PerkinElmer 2400 Series II CHNS/O Organic Elemental Analyzer.

**Table 2.** Proximate and ultimate analyses of waste machine oil and birch wood.

| Proximate Analysis | Moisture | Volatiles * | Fixed Carbon | Ash |
|---|---|---|---|---|
| Waste machine oil, wt% | 5% | - | - | 0.7% |
| Birch wood, wt% | 10–30% | 78.0% | 21.5% | 0.5% |
| Ultimate analysis | C | H | N | O |
| Waste machine oil, wt% | 85.3 | 10.9 | 0.1 | 0.4 |
| Birch wood, wt% | 48.7 | 6.4 | 0.1 | 44.5 |

* Conditions of pyrolysis: Heating to 600 °C at 20 °C/min, 30 min holding, natural cooling.

In this study, the natural gas–oxygen mixture was not diluted with steam because the detonation products themselves contained enough steam. As shown later in this paper, the equilibrium concentration of $H_2O$ in the detonation products of methane–oxygen mixture with $\Phi = 1$, expanded to 0.1 MPa is about 65%vol. All experiments are made at a fixed frequency of PD gun operation $f = 1$ Hz. To check the reproducibility of results in terms of average values of component volume fractions and run-to-run scatter, each experiment was repeated at least 3 times.

## 3. Results

### 3.1. Natural Gas Conversion

Experiments on natural gas conversion were performed with natural gas–oxygen mixtures at $0.95 \leq \Phi \leq 1.8$. Experiments with $\Phi > 1$ corresponded to the *autothermal* high-temperature $H_2O/CO_2$-assisted conversion of natural gas in the PD gun, while the flow reactor played the role of a "cooler" used to freeze the composition of detonation products due to their rapid PD gun-to-reactor volume expansion to $P = 0.1$ MPa. In experiments, the gasifier was preliminarily brought to a steady-state mode.

Figure 3 shows examples of the time histories of the syngas temperature measured by the thermocouple installed in the outlet tube downstream of the separator device (see Figure 1a,b) in two long-duration (up to 60 min) experiments with the 100- and 40-dm$^3$ reactors. In the experiments under consideration, the values of $\Phi$ were changed stepwise to cover the entire range from 0.95 to 1.8. As seen, the corresponding mean syngas temperatures reached 600–620 K and 950–1000 K, which is generally in line with expectations.

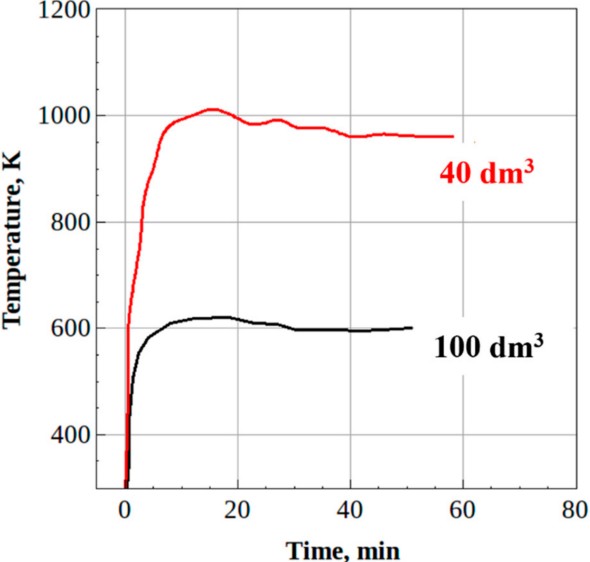

**Figure 3.** Time histories of the syngas temperature for the 100- and 40-dm$^3$ reactors with a PD gun operating at $0.95 \leq \Phi \leq 1.8$, $f = 1$ Hz, and $P \approx 0.1$ MPa.

The obtained temperature difference comes from the larger ratio of the PD tube volume to reactor volume: in the 40-dm$^3$ reactor, hot products of the new cycle were diluted in a smaller volume of expanded products of the previous cycle, as compared to the 100-dm$^3$ reactor. Due to very high velocities of incoming detonation products (over 1 km/s), the shape of reactor (spherical or cylindrical) does not play an important role. Moreover, according to [31], the wall temperature and therefore the wall surface area do not play a significant role in the time history of mean temperature. Note that the measured syngas temperatures are several hundred degrees lower than the calculated time-averaged mean temperatures of the USS inside the reactors. This difference is explained by the effect of the separator device, which increases heat loss (see insert in Figure 1b). It is worth noting that at $f$ = 1 Hz the local instantaneous maximum temperature inside the 100-dm$^3$ reactor during the first quarter of the cycle (~0.25 s) is very high ($1500 \leq T_{max} \leq 2300$ K) [31], while in the second quarter of the cycle it drops to $1200 \leq T_{max} \leq 1500$ K, and further drops below 1200 K in the second half of the cycle.

Large circles and small squares in Figure 4 show the steady-state compositions of the product syngas at different values of $\Phi$ measured in the 100- and 40-dm$^3$ reactors, respectively. The steady-state fractions of $H_2$, CO, $CO_2$, and $O_2$ are presented in different colors. The curves of the corresponding colors in Figure 4 show the results of thermodynamic calculations ($P$–$T$ problem) for the frozen composition of detonation products of methane–oxygen mixtures rapidly expanded to $P$ = 0.1 MPa, assuming that the composition-freezing temperature is 2200 K, like in [31]. The measured syngas composition is seen to be independent of the reactor volume and corresponds well with the frozen composition. This indicates that natural gas is converted in the detonation waves during their propagation along the PD gun, rather than in the flow reactor, where the mean temperature is essentially below 2200 K. The fraction of $CH_4$ was zero, i.e., the PD gun technology provided complete (100%) natural gas conversion. Thus, as compared to the mature syngas production process via natural gas reforming by high-temperature steam (1000–1300 K), the PD gun technology has several advantages: 100% conversion, lack of catalysts, normal rather than elevated (up to 2.5 MPa) operation pressure, and conventional structural materials.

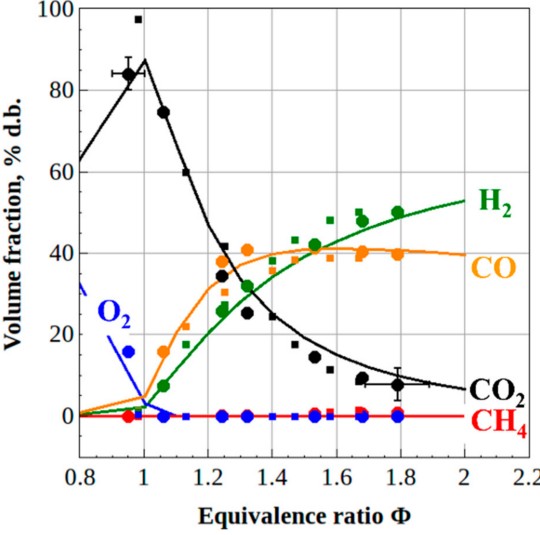

**Figure 4.** Measured (symbols) and calculated (curves) volume fractions of cooled dry detonation products of natural gas–oxygen mixture as a function of $\Phi$ for a 100-dm$^3$ reactor (circles) and 40-dm$^3$ reactor (squares) at $f$ = 1 Hz, $P \approx 0.1$ MPa. Calculations were made for the composition freezing temperature 2200 K. Bars show measurement errors.

At $\Phi \leq 1$, the detonation products were mainly composed of $H_2O$ and $CO_2$ with nearly zero fractions of $H_2$ and CO and nonzero fraction of $O_2$, which attained 15%vol dry basis (d.b.) for the 100-dm$^3$ reactor at $\Phi$ = 0.95. At $\Phi > 1$, the fraction of $O_2$ was zero,

the fraction of $H_2$ gradually increased with $\Phi$, while the fraction of CO reached a constant value of ~40%vol d.b., and the fraction of $CO_2$ decreased. At $\Phi = 1.8$, detonation products mainly contained $H_2$ (~50%vol d.b.), CO (~40%vol d.b.), and $CO_2$ (~10%vol d.b.). The maximum scatter in the measured fractions of $H_2$ and CO was ±4%. For both reactors, the $H_2$/CO and $CO_2$/CO ratios in the syngas were ~1.25 and ~0.25, respectively.

### 3.2. Liquid/Solid Waste Gasification

Experiments on liquid/solid waste gasification were performed with natural gas–oxygen mixtures with $\Phi = 1.10$ (100-dm$^3$ reactor) and 1.15 (40-dm$^3$ reactor). These values of $\Phi$ were chosen because free oxygen was absent in detonation products (see Figure 4).

Liquid waste was continuously fed by spray injectors into the PD gun. Due to aerodynamic fragmentation in PD waves, liquid waste entered the flow reactors in the form of very fine mist ~10 μm in size despite the mean droplet diameter in a free spray measured by the slide sampling method [39] being about 100 μm. The mass flow rate of liquid varied from $G = 0$ to 14 g/s. Experiment duration reached 10–15 min.

A batch of solid waste weighing 2–3 kg was loaded into a flow reactor through the lower flange. In experiments, the initial moisture content of sawdust ranged from 10 to 30%wt. Initial sawdust particle sizes ranged from 1 to 9 mm. Experiment duration reached 15–20 min.

#### 3.2.1. Liquid Waste

Figure 5 compares the time histories of the syngas temperature in the outlet tube for the 100- and 40-dm$^3$ reactors. The time point 0 corresponds to the start of liquid waste supply to the reactor with $G = 2$ g/s (100-dm$^3$ reactor) and 4 g/s (40-dm$^3$ reactor). As seen, for both reactors, there is a temperature drop of 80–90 K after the start of waste supply due to the phase transition and endothermic gasification reactions (from 750 to 660 K in the 100-dm$^3$ reactor and from 1020 to 940 K in the 40-dm$^3$ reactor) followed by the period of steady-state gasification. Note that the product syngas temperature should not be treated as the gasification temperature, as gasification reactions occurred inside the reactors at local instantaneous temperatures of the USS generated by the PD gun. As was mentioned earlier in this paper, at $f = 1$ Hz the local instantaneous maximum temperature of USS inside reactors during the first quarter of the cycle was as high as 1500–2300 K.

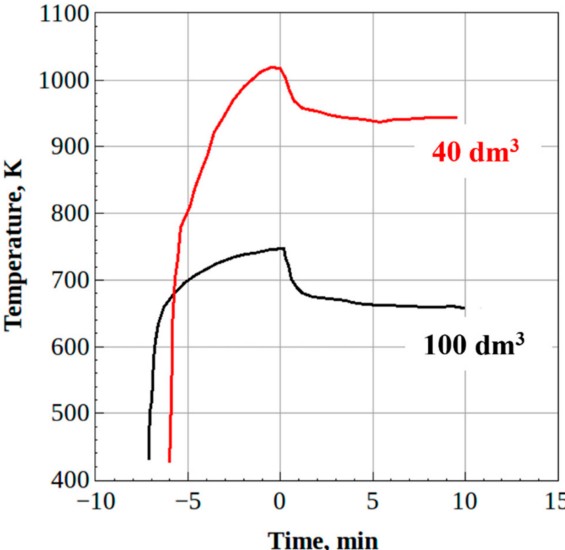

**Figure 5.** Time histories of the syngas temperature in the outlet tube for the 100- and 40-dm$^3$ reactors with the PD gun operating at $f = 1$ Hz. Time point 0 corresponds to the start of waste oil supply to the reactor with $G = 2$ g/s (100-dm$^3$ reactor) and 4 g/s (40-dm$^3$ reactor); $P \approx 0.1$ MPa.

Filled and empty symbols in Figure 6 show steady-state volume fractions of $H_2$, CO, $CO_2$, and $CH_4$ in the product syngas measured in the 100- and 40-dm$^3$ reactors, respectively, as a function of the mass flow rate of liquid waste, $G$. At $G = 0$, the syngas composition corresponds to the composition of cooled dry detonation products at $\Phi = 1.10$ and $\Phi = 1.15$. The volume fractions of $H_2$, CO, $CO_2$, and $CH_4$ in the syngas at $G > 0$ differ from their values at $G = 0$, even at small values of $G$ when the product syngas is diluted by detonation products. This is obviously a consequence of gasification reactions.

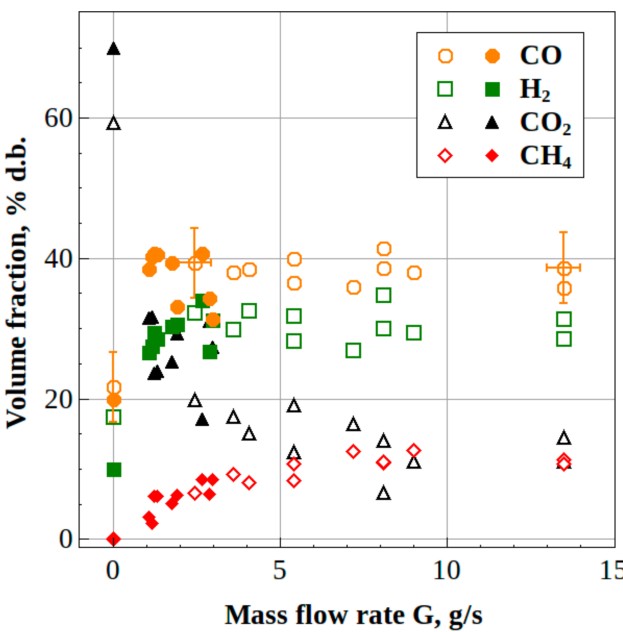

**Figure 6.** Measured compositions of cooled dry syngas as a function of $G$ ($f$ = 1 Hz, $P \approx 0.1$ MPa) obtained by liquid waste gasification in a 100-dm$^3$ (filled symbols) and 40-dm$^3$ (empty symbols) reactors. Bars show measurement errors.

In the 100-dm$^3$ reactor, the steady-state fractions of $H_2$, CO, and $CH_4$ increased from 10, 20, and 0%vol d.b. at $G = 0$ to 30–35, 30–40, and 6–10%vol d.b., at $G = 3$ g/s, respectively, while the fraction of $CO_2$ decreased from 70 to 20–30%vol d.b. At $G > 3$ g/s, some oil residue was detected in the exhaust system. This means that at $G > 3$ g/s, experimental conditions in the 100-dm$^3$ reactor at a pulse frequency of 1 Hz do not provide complete oil gasification. In the 40-dm$^3$ reactor, the steady-state fractions of $H_2$, CO, and $CH_4$ increased from 17, 22, and 0%vol d.b. at $G = 0$ to 30–35, 35–42, and 10–12%vol d.b. at $G = 13.5$ g/s, respectively, while the fraction of $CO_2$ decreased from 60 to 7–10%vol d.b. Interestingly, when the 100-dm$^3$ reactor was replaced by the 40-dm$^3$ reactor, oil residue in the exhaust system was detected at $G > 13.5$ g/s rather than at $G > 3$ g/s, thus indicating some better conditions for feedstock gasification in the 40-dm$^3$ reactor caused by the elevated local instantaneous temperature and time-averaged mean temperature of USS. In the liquid waste gasification experiments, no condensed-phase by-products like tar and char, as well as no mineral residues, were systematically detected. This was likely caused by a relatively large surface area of manifolds attached downstream from the reactor which could accumulate condensed-phase microparticles.

The maximum scatter in the measured values of steady-state fractions of $H_2$ and CO was ±5%vol for both reactors. The total fraction of combustible gases ($H_2$, CO, and $CH_4$) in the syngas increased from 30%vol d.b. at $G = 0$ to 75–89%vol d.b. at $G = 3$ g/s for the 100-dm$^3$ reactor and from 39%vol d.b to 75–89%vol d.b. at $G = 13.5$ g/s for the 40-dm$^3$ reactor. The steady-state $H_2$/CO and $CO_2$/CO ratios in the syngas were approximately 0.8 and 0.5 for the 100-dm$^3$ reactor vs. 0.9 and 0.2 for the 40-dm$^3$ reactor. These numbers indicate that the product syngas quality is somewhat higher and syngas contamination

with $CO_2$ is somewhat lower for the 40-$dm^3$ reactor, which is obviously the consequence of higher gasification temperature in the 40-$dm^3$ reactor.

3.2.2. Solid Waste

Experiments with solid waste were performed using a batch feed scheme. A batch of waste was loaded in the flow reactor and gasified over some period. At a certain point in time, syngas composition temporarily reached a steady-state level and after a certain period of gasifier operation, syngas composition began to change again, indicating the completion of the gasification process due to the lack of feedstock.

Figure 7 compares the time histories of temperature in the outlet tube for the 100- and 40-$dm^3$ reactor. Time point 0 corresponds to the start of gasification experiments in the cold reactor loaded with a fixed 2-kg batch of wood sawdust. As seen, the product gas temperature gradually grows to ~700 K for the 100-$dm^3$ reactor and to ~900 K for the 40-$dm^3$ reactor until all the feedstock is gasified (see arrows in Figure 7), and thereafter jumps to ~750 K and ~1000 K, respectively, when the reactors get empty. Again, it must be noted that the gasification of solid waste in the experiments under consideration proceeds at local instantaneous temperatures of the USS significantly exceeding the product syngas temperature measured by the thermocouple. In both reactors, the gasification time of identical feedstock batches was nearly the same: about 5–7 min for a 2-kg batch load, thus indicating a sort of compensation effect of higher gasification temperature and lower volume of the 40-$dm^3$ reactor as compared to the 100-$dm^3$ reactor.

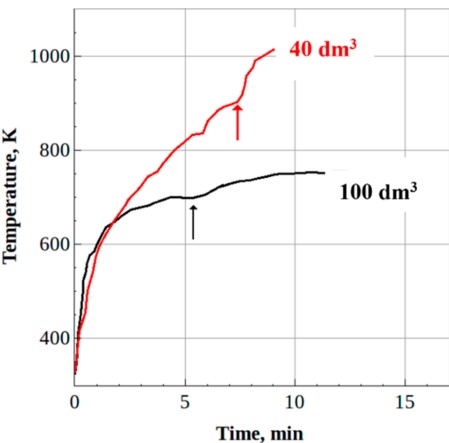

**Figure 7.** Time histories of syngas temperature in the outlet tube for the 100- and 40-$dm^3$ reactors with a PD gun operating at $f = 1$ Hz. Time point 0 corresponds to the start of gasification experiments with the cold reactor loaded by a 2-kg batch of wood sawdust.

Figure 8 shows the measured time history of the cooled dry syngas composition corresponding to the gasification experiment of Figure 7 in the 40-$dm^3$ reactor. There is a certain time delay between the readings of thermocouple and gas analyzer caused by their different response times. The steady-state composition of the product syngas is established by about 5 min after starting the gasification experiment and is maintained for about 3 min up to completion of the gasification process.

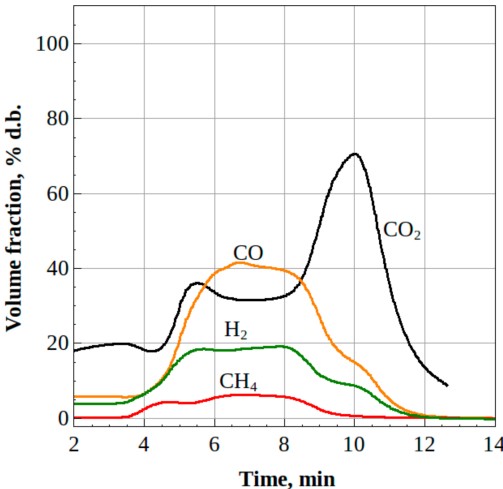

**Figure 8.** Measured time history of the cooled dry syngas composition during gasification of a 2-kg batch of wood sawdust at $f = 1$ Hz and $P \approx 0.1$ MPa in a 40-dm$^3$ reactor.

Figure 9 shows steady-state volume fractions of $H_2$, CO, $CO_2$, and $CH_4$ in the product syngas measured in the 100- and 40-dm$^3$ reactors. In both reactors, the steady-state fractions of $H_2$, CO, $CO_2$, and $CH_4$ attained values of 17–19, 39–41, 31–34, and 3–6%vol d.b., respectively. Interestingly, the steady-state composition of the product syngas was reasonably reproduced from experiment to experiment, not only with the identical batch loads but also with different batch loads (2 or 3 kg feedstock). Moreover, the steady-state compositions of the product syngas were similar for sawdust, with different dominating particle sizes ranging from 1 to 9 mm and different moisture ranging from 10 to 30%wt. This apparently happened because strong shock waves and supersonic USS jets facilitated rapid primary fragmentation of the packed pile of wet sawdust particles, secondary fragmentation of particles, fast moisture vaporization, and short residence times of gases in the reactors.

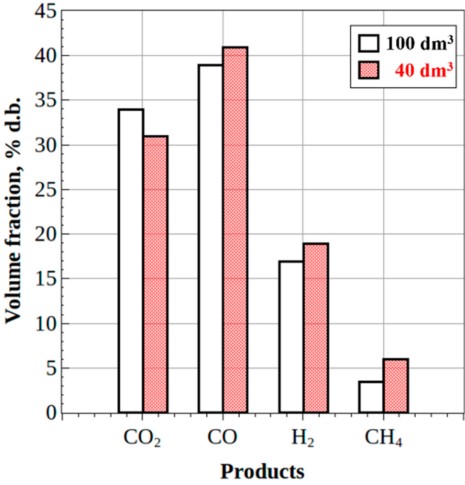

**Figure 9.** Measured steady-state compositions of cooled dry syngas obtained by gasification of wood sawdust at $f = 1$ Hz and $P \approx 0.1$ MPa in 100- and 40-dm$^3$ reactors.

The maximum scatter in the measured values of steady-state fractions of $H_2$ and CO was ±5%vol. The steady-state $H_2$/CO and $CO_2$/CO ratios in the syngas were ~0.5 and 0.8. The total fraction of combustible gases in the syngas ($H_2$, CO, and $CH_4$) increased from 30 to 65%vol d.b. for the 100-dm$^3$ reactor and from 40 to 65%vol d.b. for the 40-dm$^3$ reactor. In the solid waste gasification experiments, no condensed-phase by-products like tar and char, as well as no mineral residues, were systematically detected. Similar to the liquid

waste, this was probably caused by a relatively large surface area of manifolds attached downstream from the reactor which could accumulate condensed-phase microparticles.

## 4. Discussion

Large and small empty symbols in Figure 10 show the steady-state compositions of the product syngas obtained in experiments on natural gas conversion and liquid/solid waste gasification under the same conditions ($f$ = 1 Hz, $P \approx 0.1$ MPa) in the 100- and 40-dm$^3$ reactors, respectively. The steady-state volume fractions of $H_2$, $CO$, and $CH_4$ are presented in different colors as a function of the steady-state $CO_2$ volume fraction in the product syngas. For the sake of generality, some literature data on syngas composition obtained by high-temperature plasma-assisted techniques of organic waste gasification [19,22] are also presented in Figure 10 by large, filled symbols of the corresponding colors. In [19,22], the plasma torch with hybrid water/gas stabilization of arc was used for steam/$CO_2$ gasification of pyrolysis oil, sawdust, and refuse-derived fuel (RDF) at a mean process temperature of 1150–1350 K and pressure of 0.1 MPa, which is close to the process mean temperature and pressure in the current experiments. For pyrolysis oil, the fractions of $H_2$, $CO$, and $CH_4$ in the syngas were 58, 33, and 4.9%, respectively, at $CO_2$ fraction of 4% [19]. For sawdust, the fractions of $H_2$, $CO$, and $CH_4$ in the syngas were 42–43, 44–49, and 1.3–1.7%, respectively, at a $CO_2$ fraction of 4.7–7.2% [22]. For RDF, containing 24%wt wood and paper, 47%wt plastics, 18%wt fines, and 10%wt textiles, the fractions of $H_2$, $CO$, and $CH_4$ in the syngas were 41–50, 44–41, and 5–3%, respectively, at $CO_2$ fraction of 9.4–6.5% [19]. As seen from Figure 10, with a decrease in the fraction of $CO_2$ in the syngas, the fractions of $H_2$ and $CO$ increase, with $H_2$ tending to 60%vol d.b. and $CO$ reaching a plateau at ~40%vol d.b. Interestingly, the composition of the syngas in terms of $H_2$ and $CO$ volume fractions looks almost independent of the reactor volume and the type of organic feedstock. Additionally, the data of [19,22] satisfactorily fit the data obtained herein.

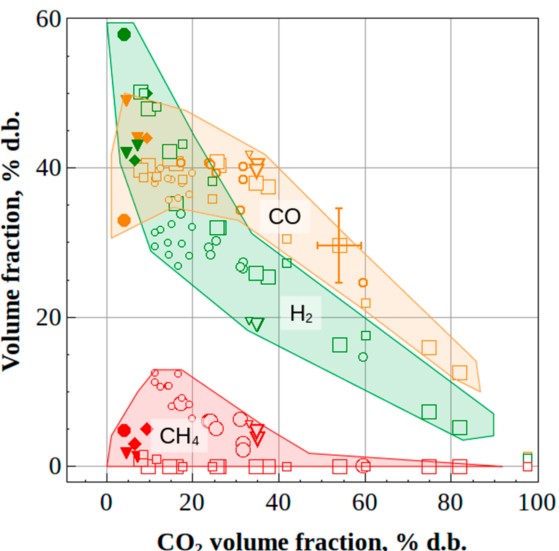

**Figure 10.** Comparison of syngas compositions in experiments on natural gas conversion (squares), and liquid (circles) and solid (triangles) waste gasification at $f$ = 1 Hz and $P \approx 0.1$ MPa. Large empty symbols correspond to the 100-dm$^3$ reactor, small empty symbols correspond to the 40-dm$^3$ reactor, large filled symbols correspond to plasma reactors [19,22], diamonds correspond to RDF gasification [19].

To explain the similarity of syngas compositions for different reactor volumes (100 and 40 dm$^3$), different types of organic feedstocks (natural gas, waste oil, sawdust), and different properties of solid waste (batch load, particle size, moisture), let us consider relevant gasification reactions (1) and (2). Equilibrium calculations of organic waste gasification [40] indicate that at a temperature of ~900 K, oxygen and carbon exist as $CO_2$, tar, and char;

at a temperature exceeding ~1200 K, available oxygen preferably reacts with carbon to form $CO_2$ and CO rather than with hydrogen to form $H_2O$, and $CO_2$ breaks down to CO in presence of carbon; and at a temperature exceeding ~1800 K, tar and char are converted to syngas composed of $H_2$ and CO. The fact that condensed-phase by-products like tar and char and mineral residues were not detected in the experiments on liquid/solid waste gasification could be explained by two reasons. Firstly, the feedstocks contained small amounts of ash, which could be accumulated in the form of condensed-phase microparticles at internal surfaces of manifolds attached downstream from the reactor. Secondly, the local instantaneous USS temperatures generated by the PD gun were very high (above 2000 K) and the residence time of feedstocks in the reactors were sufficiently long (up to 15 s [31,36]). According to Section 2.3, waste oil and natural gas contained comparable amounts of carbon (~85 and ~95%wt, respectively), while wood sawdust contained ~45%wt oxygen and ~49%wt carbon. It is not then surprising that syngas compositions for waste oil and natural gas in Figure 10 are close to each other. Regarding wood sawdust, at high local instantaneous USS temperatures the oxygen available in wood mostly reacted with carbon to form CO, and the volume fraction of CO in the sawdust gasification products was close to that in the gasification products of waste oil and natural gas. Wood sawdust properties such as batch load, moisture, and particle size distribution appeared to be much less significant compared to other gasification technologies, presumably because strong shock waves and supersonic USS jets facilitated rapid primary fragmentation of the packed piles of wet sawdust particles, secondary fragmentation of particles, fast moisture vaporization, and short residence times of gases in the reactors.

Following [31], it is expected that the increase in the operation frequency of the PD gun from 1 to 5 Hz, other conditions being equal, will lead to a significant increase in the mean rates of reactions (1) and (2), while the volume fractions of $H_2$ and CO in the product syngas will tend to 60 and 40%vol, respectively, and the contents of $CH_4$ and $CO_2$ will tend to zero. One can readily use available downstream techniques to increase the content of hydrogen in the product syngas. Thus, the PD gun technology has several attractive features for organic waste gasification, as (i) it allows atmospheric pressure operation at very high gasification temperatures obtained by detonating, e.g., natural gas as a starting fuel and a part of produced syngas (about 10% [31]) as a process fuel, while the energy consumption for detonation ignition is negligible; (ii) it can potentially ensure the complete conversion of gas, liquid, and solid wastes into useful products—syngas, consisting exclusively of hydrogen and carbon monoxide, fine particles of mineral residues, consisting of safe simple oxides, as well as aqueous solutions of simple acids like HCl, HF, $H_2S$, etc., and ammonia $NH_3$; (iii) it can be implemented using conventional structural materials; (iv) it can be scaled-up by applying multiple PD guns of the same power or PD guns of higher power designed based on the well-known scaling laws inherent in detonations; and (v) it is economically beneficial, as the ideal energy gain for syngas production from dry wood sawdust is about 4.6 [31].

### 5. Conclusions

In this work, the pulsed detonation gun technology was applied for the *autothermal* high-temperature conversion of natural gas and atmospheric-pressure oxygen-free *allothermal* gasification of waste machine oil and birch wood sawdust by detonation-born ultra-superheated steam using a pulse frequency of 1 Hz and two flow reactors of essentially different volume: 100 and 40 $dm^3$. The following main results were obtained:

(1) Complete conversion of natural gas to product syngas was obtained with $H_2$/CO and $CO_2$/CO ratios equal to 1.25 and 0.25, irrespective of the reactor volume. As compared to the mature syngas production process via natural gas reforming by high-temperature steam (1000–1300 K), the PD gun technology has several advantages: 100% conversion, lack of catalysts, normal operation pressure, and conventional construction materials.

(2) The steady-state $H_2/CO$ and $CO_2/CO$ ratios in the syngas produced from waste machine oil were 0.8 and 0.5 for the 100-$dm^3$ reactor and 0.9 and 0.2 for the 40-$dm^3$ reactor, respectively, thus indicating the improvement in syngas quality. Moreover, the mass flow rate of feedstock in the 40-$dm^3$ reactor was increased by a factor of over 4, as compared to the 100-$dm^3$ reactor.

(3) The steady-state $H_2/CO$ and $CO_2/CO$ ratios in the syngas produced from the fixed weight (2 kg) batches of wood sawdust were 0.5 and 0.8 for both reactors, while the gasification time of the batches was nearly the same.

(4) Wood sawdust properties such as moisture and particle size distribution appeared to be less significant compared to other gasification technologies presumably because strong shock waves and supersonic USS jets facilitated rapid primary fragmentation of the packed piles of wet sawdust particles, secondary fragmentation of particles, fast moisture vaporization, and short residence times of gases in the reactors.

(5) The measured $H_2$ vs. $CO_2$ and CO vs. $CO_2$ dependences for the syngas produced by the *autothermal* high-temperature conversion of natural gas and atmospheric-pressure oxygen-free *allothermal* gasification of liquid/solid organic wastes by detonation-born ultra-superheated steam were shown to be almost independent of the feedstock and reactor volume, which was caused by high local instantaneous gasification temperatures in the reactors.

(6) The increase in the operation frequency of the pulsed detonation gun from 1 to 5 Hz, other conditions being equal, is expected to lead to a drastic increase in the mean rates of gasification reactions, while the fractions of $H_2$ and CO in the product syngas must tend to ~60 and ~40%vol, respectively, and the contents of $CH_4$ and $CO_2$ must tend to zero.

**Author Contributions:** S.M.F.: Conceptualization; methodology, writing original draft & editing; supervision; funding acquisition; project administration; V.A.S.: methodology, investigation, data curation; I.A.S.: investigation; A.S.S.: investigation; I.O.S.: investigation; V.S.A.: investigation; K.A.A.: investigation; F.S.F.: investigation. All authors have read and agreed to the published version of the manuscript.

**Funding:** This research received no external funding.

**Institutional Review Board Statement:** Not applicable.

**Informed Consent Statement:** Not applicable.

**Data Availability Statement:** The data presented in this study are available on request from the corresponding author.

**Conflicts of Interest:** The authors declare no conflict of interest.

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
