# Peer review of "Natural Gas Conversion and Organic Waste Gasification by Detonation-Born Ultra-Superheated Steam: Effect of Reactor Volume"

_2673-3994, doi:10.3390/fuels3030024_

Round 1
Reviewer 1 Report
This work proposes pulse detonation (PD) gun technology as a tool for natural gas reforming and liquid and solid wastes gasification to obtain syngas. The work presents some novelty in the final use of the technique, so could be interesting. But there are some points to be cleared before being published.
Firstly, after reading the manuscript I still have a huge doubt. Which is exactly the function of the newly proposed technology (PD gun) in the gasification process?. As this is not clearly explained in the work. Is it useful to enhance and maintain the required operation temperature, crunch/mix feedstocks, provide adequate reaction environment…?. This should be explained.
Additionally, I think the expected chemical reactions should be summarized in the introduction, it may make easier to follow the works nature.
Regarding operative questions. Why is the cylindrical flow reactor not equipped with a cooling jacket? Is it due to any operational issue related to its size? As it seems easier to use a cooling jacket in a cylindrical device rather than in a spherical one, so it will make operation control easier.
It is stated that the maximum temperature reached into the reactor is close to 1500-2300K. Is it not available a more accurate measure of the internal reactor temperature during reforming process?
About the concept of the work itself, it also has to be said that methane/natural gas reforming or even solid/liquid gasification to produce syngas or, more commonly now, pure hydrogen is a well known process for which there are available mature technologies in literature, such as regular gasification or steam methane reforming. I think the main weak spot of this work is it lacks a deep comparison between these traditional techniques and the one proposed by authors. Which are the main advantages and flaws of the proposed technique when compared to the most commonly used for this aim?.
Regarding up-scale and future use of this technique for this aim, do authors think this technique is feasible to be up-scaled for mass production of syngas?. From that point of view, I miss a broad economy balance. As PD gun seems to be a huge energy consuming device, which maybe expensive to use, especially when considering production of a “cheap” feedstock such as syngas. Observing results, it can be concluded that even at the best reaction conditions achieved, there is obtained a fraction of CO2 what impoverishes the product syngas. Could this process be coupled to other techniques to produce a more sophisticated and cleaner gaseous product (i.e. higher H2 ratio without CO2), that may have a higher added value so it can compensate the energy expenditures of using this technique.
Author Response
We are grateful to the reviewer for valuable comments. We made our best to follow all the comments. All changes in the revised manuscript are marked in yellow.
This work proposes pulse detonation (PD) gun technology as a tool for natural gas reforming and liquid and solid wastes gasification to obtain syngas. The work presents some novelty in the final use of the technique, so could be interesting. But there are some points to be cleared before being published.
Firstly, after reading the manuscript I still have a huge doubt. Which is exactly the function of the newly proposed technology (PD gun) in the gasification process?. As this is not clearly explained in the work. Is it useful to enhance and maintain the required operation temperature, crunch/mix feedstocks, provide adequate reaction environment…?. This should be explained.
To follow this comment, we have supplemented Introduction with the sentence:
“Currently, 1 Hz is the maximum frequency of the available PD gun operating on natural gas–oxygen–steam mixture. This frequency is limited by the oxygen flow rate provided by the existing liquid oxygen cryocylinder. It is implied that the PD gun technology will allow maintaining highly reactive environment in a gasifier due to very high operation temperature, intense detonation-induced crunching and mixing of feedstock, and high-speed convective flows. These distinctive and unique features of the technology can potentially ensure the complete conversion of gas, liquid, and solid wastes into useful products – syngas, consisting exclusively of hydrogen and carbon monoxide, fine particles of mineral residues, consisting of safe simple oxides, as well as aqueous solutions of simple acids like HCl, HF, H2S, etc., and ammonia NH3.”
Additionally, I think the expected chemical reactions should be summarized in the introduction, it may make easier to follow the works nature.
To follow this comment, we moved the extended list of expected chemical reactions from the Discussion section to the introduction.
Regarding operative questions. Why is the cylindrical flow reactor not equipped with a cooling jacket? Is it due to any operational issue related to its size? As it seems easier to use a cooling jacket in a cylindrical device rather than in a spherical one, so it will make operation control easier.
The simplest way of increasing the operation temperature in the reactor is to increase the operation frequency, i.e., by detonating larger volumes of natural gas -oxygen mixture. However, available liquid oxygen cryocylinder cannot currently allow for a larger oxygen flow rate. Therefore, for increasing the operation temperature at a fixed frequency of detonation pulses we reduced the reactor volume from 100 to 40 liters and purposefully refused to cool the walls. Of course, all practical reactors are better to design water cooled with the use of hot water (steam) in the waste gasification process. To follow this comment, we have replaced the sentence: “As for the cylindrical flow reactor of Fig. 1b, it is not water cooled and can warm up to a red glow (see insert in Fig. 1b)” by
“As for the cylindrical flow reactor of Fig. 1b, it is purposefully not water cooled and can warm up to a red glow (see insert in Fig. 1b) for increasing the operation temperature.”
It is stated that the maximum temperature reached into the reactor is close to 1500-2300K. Is it not available a more accurate measure of the internal reactor temperature during reforming process?
The mentioned maximum temperature in the reactor are the results of 3D calculations. Since the measurements of rapidly changing local instantaneous temperatures inside the reactor are very problematic, we measured the mean gas temperature downstream of the reactor exit, where the flow conditions are more stable. To follow this comment, we have added a new Fig. 2 and several sentence to the subsection 2.2:
“To demonstrate the temperature patterns and the way of its change in the reactor, Figure 2 shows the calculated snapshots of gas temperature during a time interval of 0.1 s after one of multiple detonation shots in the 100-dm3 spherical reactor partly filled with 1-mm-size waste particles. The calculation is made using the approach discussed in [31, 36]. As seen, the local instantaneous temperatures inside the reactor change very rapidly and attain the values between 1500 and 2500 K. Since the measurements of rapidly changing local instantaneous temperatures inside the reactor are very problematic, the mean syngas temperature is measured downstream of the separator device by the thermocouple installed in the outlet tube.”
About the concept of the work itself, it also has to be said that methane/natural gas reforming or even solid/liquid gasification to produce syngas or, more commonly now, pure hydrogen is a well known process for which there are available mature technologies in literature, such as regular gasification or steam methane reforming. I think the main weak spot of this work is it lacks a deep comparison between these traditional techniques and the one proposed by authors. Which are the main advantages and flaws of the proposed technique when compared to the most commonly used for this aim?.
To follow this comment, we have added two sentences: one to the subsection 3.1 :
“Thus, as compared to the mature syngas production process via natural gas reforming by high-temperature steam (1000–1300 K), the PD gun technology has several advantages: 100% conversion, lack of catalysts, normal rather than elevated (up to 2.5 MPa) operation pressure, and conventional structural materials,”
and another to the Conclusions:
“As compared to the mature syngas production process via natural gas reforming by high-temperature steam (1000–1300 K), the PD gun technology has several advantages: 100% conversion, lack of catalysts, normal operation pressure, and conventional construction materials.”
As for the comparison of the PD gun technology with conventional gasification processes, we have provided many references for low-temperature and high-temperature gasification technologies in the Introduction section. Nevertheless, to follow this comment, we have added several more sentences at the end of the Discussion section:
“One can readily use available downstream techniques to increase the content of hydrogen in the product syngas. Thus, the PD gun technology has several attractive features for organic waste gasification, as (i) it allows atmospheric pressure operation at very high gasification temperatures obtained by detonating, e.g., natural gas as a starting fuel and a part of produced syngas (about 10% [31]) as a process fuel, while the energy consumption for detonation ignition is negligible; (ii) it can potentially ensure the complete conversion of gas, liquid, and solid wastes into useful products – syngas, consisting exclusively of hydrogen and carbon monoxide, fine particles of mineral residues, consisting of safe simple oxides, as well as aqueous solutions of simple acids like HCl, HF, H2S, etc., and ammonia NH3; (iii) it can be implemented using conventional structural materials; (iv) it can be scaled-up by applying multiple PD guns of the same power or PD guns of high power designed based on the well-known scaling laws inherent in detonations; and (v) it is economically beneficial, as the ideal energy gain for syngas production from dry wood sawdust is about 4.6 [31].”
Regarding up-scale and future use of this technique for this aim, do authors think this technique is feasible to be up-scaled for mass production of syngas?. From that point of view, I miss a broad economy balance. As PD gun seems to be a huge energy consuming device, which maybe expensive to use, especially when considering production of a “cheap” feedstock such as syngas. Observing results, it can be concluded that even at the best reaction conditions achieved, there is obtained a fraction of CO2 what impoverishes the product syngas. Could this process be coupled to other techniques to produce a more sophisticated and cleaner gaseous product (i.e. higher H2 ratio without CO2), that may have a higher added value so it can compensate the energy expenditures of using this technique.
Mass and energy balances were estimated in [31] and we mention it in the Introduction:
“According to estimates [31], about 10% (ideally) of the syngas is needed for replacing natural gas for USS production in the PD gun, whereas the rest can be used for downstream needs.”
Nevertheless, we have placed the statements on the balance and up-scaling of the PD gun technology to the end of the Discussion section. Note that the PD gun is a device of very low energy consumption, and we have pointed it out additionally in the text (see subsection 2.1: “Since the PD gun operates on deflagration-to-detonation transition, it utilizes very low energy for detonation initiation.” As for the residual CO2 in the product syngas, it will be diminished by increasing the operation temperature, as mentioned at the end of the Discussion section. As said above in our response to the reviewer, at this stage of research, we could not increase the operation frequency since the available liquid oxygen cryocylinder did not allow for a larger oxygen flow rate. Regarding the higher added value of the produced syngas, we have added another sentence at the end of the Discussion section: “One can readily use available downstream techniques to increase the content of hydrogen in the product syngas.”

Reviewer 2 Report
Dear Authors,
The work contains errors, omissions and ambiguities.
1. There is no formulation of the energy balance for both reactors (the shapes and structures of the reactors are different). What share in the energy balance is attributed to the shock wave? The temperature distribution is important here.
2. What is the efficiency of the process and at what stages of the process should it be considered?
3. The explanation regarding the obtained high temperature differences in both reactors should be clarified - show in detail where the differences come from. Perhaps there is an effect of the detonation wave (different in both cases).
4. The authors sometimes use SI units, eg temperature K, and sometimes the old units, eg l instead of dm3. Always use SI units.
When reading the presented manuscript, one gets the impression that it is an advertisement of the authors' own works. Of course, you can, and should, link to your own articles if this is to expand the knowledge covered in the manuscript. However, this cannot mean that the reader has to refer to the authors' works in order to get acquainted with what is included in the presented work.
Detailed remarks:
Chapter 2.1 Technology
Fig. 1.
Why was the spherical shape adopted for the 100 dm3 reactor, and the cylindrical shape for the 40 dm3 reactor? What is the influence of the shape of the reactor on the course of the process and temperature?
Chapter 2.3 Materials
Table 2
The procedures (standards) according to which the analyzes were performed, were not specified.
The elemental analyzer used to determine the CHN was not specified.
Elemental composition analysis: Why were S and Cl elements not determined, but only C, H, N and O calculated? Moreover, the sum of the shares of these elements should be 100%, and the sum of the values given by the authors is lower (assuming that it concerns ashless dry mass). If we assume that the analysis concerns dry ash weight, the sum is also lower for waste machine oil, and for birch wood it is higher than 100%. These are significant analytical errors. In addition, information should be provided whether this analysis concerns the elemental composition of the combustible substance of the waste only, or the waste in a dry or working state and ashes.
To be corrected in the fifth column: it is "Ah", it should be "Ash".
Fig. 3
Please refer to the value in the area 0.8 <Φ> 1 (in the context of the process).
Chapter 3.2.1 Liquid waste and Chapter 3.2.2 Solid waste
How were the temperature zones in the reactors distributed: 100 dm3 and 40 dm3?
Chapter 3.2.1 Liquid waste
Quote: „……no mineral residues were detected.”
Comment: Please explain this because table 2 shows the value of ash as 0.7%.
Chapter 3.2.2 Solid waste
Quote: „……no mineral residues were detected.”
Comment: Please explain this as table 2 shows the value of ash as 0.5%.
Chapter 4. Discussion
1. 1. Quote: „……mineral residues were not detected in the experiments on liquid/solid waste gasification……..Firstly, the feedstocks contained nearly no ash.”
Comment: It's not true. The ash amounts in Table 2 show that after gasification of e.g. 2-3 kg of solid waste (chapter 3.2 Liquid / solid waste gasification), 10-15 g of ash remains, which is a traceable amount. The same applies to liguid waste containing 0.7% ash.
2. Quote: „In the absence of free oxygen, the gasification processes of liquid/solid wastes are known to include heterogeneous (endothermic) reactions:
C + H2O = CO + H2
C + CO2 = 2CO
C + 2H2 = CH4
and gas-phase reactions:
CO + H2O = CO2 + H2
CnHm + nH2O = nCO + (n+0.5m)H2
CnHm + nCO2 = 2nCO + 0.5mH2”
Comment: There are errors. The reaction C + 2H2 = CH4 is an exothermic reaction: –74,9 kJ/mol. Also the reaction CO + H2O = CO2 + H2 is an exothermic reaction: –41,2 kJ/mol.
It is unclear why the authors did not report exothermic reactions, which also occur e.g:
2C + 2H2O = CH4 + CO2 -8,79kJ/mol
CO + 3H2 = CH4+H2O -205kJ/mol
CO2 + 4H2 = CH4+ 2H2O –167,3 kJ/mol
2CO + 2H2 = CH4+ CO2 –247 kJ/mol
Best regards,
Reviewer
Author Response
We are grateful to the reviewer for valuable comments. We made our best to follow all the comments. All changes in the revised manuscript are marked in blue.
Dear Authors,
The work contains errors, omissions and ambiguities.
1. There is no formulation of the energy balance for both reactors (the shapes and structures of the reactors are different). What share in the energy balance is attributed to the shock wave? The temperature distribution is important here.
The reviewer is right: the temperature distribution is very important here. To show the instantaneous temperature distributions, we have added a new figure, Fig. 2 and several sentences to the manuscript in subsection 2.2:
“To demonstrate the temperature patterns and the way of its change in the reactor, Figure 2 shows the calculated snapshots of gas temperature during a time interval of 0.1 s after one of multiple detonation shots in the 100-dm3 spherical reactor partly filled with 1-mm-size waste particles. The calculation is made using the approach discussed in [31, 36]. As seen, the local instantaneous temperatures inside the reactor change very rapidly and attain the values between 1500 and 2500 K. Since the measurements of rapidly changing local instantaneous temperatures inside the reactor are very problematic, the mean syngas temperature is measured downstream of the separator device by the thermocouple installed in the outlet tube.”
As for the energy share in a detonation wave, we have added several sentences to the text in the Introduction section and a new reference [30] to the list of references:
“The important benefit of detonation over conventional combustion is that in addition to higher thermal energy it provides the reaction products with a considerable kinetic energy. This kinetic energy can be utilized for enhancing feedstock gasification by means of shock-induced feedstock fragmentation and efficient interphase mixing, heat, and mass transfer. The ratio of the kinetic energy of the detonation products to their enthalpy at the Chapman-Jouguet plane in the fixed set of coordinates is approximately (γ − 1)/2γ^2 [30], where γ is the ratio of specific heats. At γ ≈ 1.3, the kinetic energy share in the propagating detonation wave is about 9%.”
As for the mass and energy balances for the PD gun technology, they were estimated in [31]. In the present manuscript, we mention it in the subsection 2.2. Nevertheless, to follow this comment, we have added the sentence to the Introduction section:
“Importantly, a part of product syngas (ideally about 10% [31]) can be used for replacing natural gas (starting fuel) for USS production in the PD gun, whereas the rest can be used for downstream needs.”
2. What is the efficiency of the process and at what stages of the process should it be considered?
The thermodynamic efficiency of the detonation (Zel’dovich) cycle is known to be considerably higher than that of constant-pressure combustion cycle in terms of the useful work produced by the reaction products, other conditions being equal. The efficiency of detonation cycle is even somewhat higher than that of the constant-volume combustion cycle. The benefit of the detonation cycle is that in addition to the thermal energy it provides a considerable kinetic energy. This kinetic energy can be utilized for feedstock fragmentation, as well as efficient interphase mixing, heat and mass transfer, i.e., the processes enhancing feedstock gasification. To follow this comment, we have added several sentences in the Introduction section:
“The important benefit of detonation over conventional combustion is that in addition to higher thermal energy it provides a considerable kinetic energy. This kinetic energy can be utilized for enhancing feedstock gasification by means of shock-induced feedstock fragmentation and efficient interphase mixing, heat, and mass transfer. The ratio of the kinetic energy of the detonation products to their enthalpy at the Chapman-Jouguet plane in the fixed set of coordinates is approximately (γ − 1)/2γ^2 [30], where γ is the ratio of specific heats. At γ ≈ 1.3, the kinetic energy share in the detonation wave is about 9%.”
3. The explanation regarding the obtained high temperature differences in both reactors should be clarified - show in detail where the differences come from. Perhaps there is an effect of the detonation wave (different in both cases).
The obtained high temperature differences in both reactors come from the larger PD tube volume–to–reactor volume ratio. At a fixed frequency, this ratio is responsible for the way the reactor is refilled with hot detonation products: in the smaller-volume reactor, the hot products of the new cycle are diluted in a smaller volume of expanded products of the previous cycle. Due to very high velocities of incoming detonation products, the shape of reactor does not play an important role. Moreover, according to our calculations in [31], the wall temperature and therefore the wall surface area do not play a significant role in the time history of mean temperature. To follow this comment, we have added several sentences to the subsection 3.1:
“The obtained temperature difference comes from the larger ratio of the PD tube volume to reactor volume: in the 40-dm3 reactor, hot products of the new cycle were diluted in a smaller volume of expanded products of the previous cycle, as compared to the 100-dm3 reactor. Due to very high velocities of incoming detonation products (over 1 km/s), the shape of reactor (spherical or cylindrical) does not play an important role. Moreover, according to [31], the wall temperature and therefore the wall surface area do not play a significant role in the time history of mean temperature.”
4. The authors sometimes use SI units, eg temperature K, and sometimes the old units, eg l instead of dm3. Always use SI units.
We have changed all units to SI units both in the text and in figures.
When reading the presented manuscript, one gets the impression that it is an advertisement of the authors' own works. Of course, you can, and should, link to your own articles if this is to expand the knowledge covered in the manuscript. However, this cannot mean that the reader has to refer to the authors' works in order to get acquainted with what is included in the presented work.
The reason we must refer to our own studies is that the PD gun technology is patented and developed by us, and many of its specific features can be found only in our previous publications. So, an interested reader could find many useful details therein. We would be happy if other researchers launch similar studies.
Detailed remarks:
Chapter 2.1 Technology
Fig. 1.
Why was the spherical shape adopted for the 100 dm3 reactor, and the cylindrical shape for the 40 dm3 reactor? What is the influence of the shape of the reactor on the course of the process and temperature?
We believe that the shape (spherical or cylindrical) does not play a significant role as the incoming supersonic jets rapidly fill the whole volume, while the mean pressure is maintained nearly constant at about 0.1 MPa. The jet velocity is over 1 km/s. To follow this comment, we have added several sentences to the subsection 3.1:
“The obtained temperature difference comes from the larger ratio of the PD tube volume to reactor volume: in the 40-dm3 reactor, hot products of the new cycle were diluted in a smaller volume of expanded products of the previous cycle, as compared to the 100-dm3 reactor. Due to very high velocities of incoming detonation products (over 1 km/s), the shape of reactor (spherical or cylindrical) does not play an important role. Moreover, according to [31], the wall temperature and therefore the wall surface area do not play a significant role in the time history of mean temperature.”
Chapter 2.3 Materials
Table 2
The procedures (standards) according to which the analyzes were performed, were not specified. The elemental analyzer used to determine the CHN was not specified.
We have specified the Elemental Analyzer in the text by adding a sentence to subsection 2.3:
“The elemental analysis was made with PerkinElmer 2400 Series II CHNS/O Organic Elemental Analyzer [31].”
Elemental composition analysis: Why were S and Cl elements not determined, but only C, H, N and O calculated? Moreover, the sum of the shares of these elements should be 100%, and the sum of the values given by the authors is lower (assuming that it concerns ashless dry mass). If we assume that the analysis concerns dry ash weight, the sum is also lower for waste machine oil, and for birch wood it is higher than 100%. These are significant analytical errors. In addition, information should be provided whether this analysis concerns the elemental composition of the combustible substance of the waste only, or the waste in a dry or working state and ashes.
As for the differences addressed by the reviewer, the analysis used at this preliminary research stage did not include Si, Cl, P, and other substances like metals, whereas the amount of S was negligeable. The differences in elemental analysis are also caused by some measurement errors, which could be relatively large for some elements like O. Nevertheless, the composition of our waste oil and birch sawdust reasonably correlates with other sources, where the sum of all detected elements is close but never equal to 100% (except for the cases when O is determined by difference), see:
Waste Oil
1.Song GJ, Seo YC, Pudasainee D, Kim IT. Characteristics of gas and residues produced from electric arc pyrolysis of WLO. Waste Manag 2010;30:1230–7. https://doi.org/10.1016/j.wasman.2009.10.004 (Sum = 102%).
2.Seyed Emadeddin Alavi, Mohammad Ali Abdoli, Farhad Khorasheh, Abdolmajid Bayandori Moghaddam. Non-isothermal pyrolysis of used lubricating oil and the catalytic effect of carbon-based nanomaterials on the process performance. Journal of Thermal Analysis and Calorimetry. https://doi.org/10.1007/s10973-019-08436-w (Sum = 98.2%)
3.Haijun Guo, Fen Peng, Hairong Zhang, Lian Xiong, Shanggui Li, Can Wang, Bo Wang, Xinde Chen, Yong Chen. Production of hydrogen rich bio-oil derived syngas from co-gasification of bio-oil and waste engine oil as feedstock for lower alcohols synthesis in two-stage bed reactor. (2014). Int. J. Hydrogen Energy, http://dx.doi.org/10.1016/j.ijhydene.2014.04.008 (Sum = 100.1%)
4.Dragne, M., Pop, E., Covaliu, C. I., Matei, E., & Pîşă, I. (2019). Experiments in use of bentonite for energy recovery of used oils. E3S Web of Conferences, 112, 04009. http://dx.doi.org/10.1051/e3sconf/201911204009 (Sum = 97.7-98.7%).
Birch sawdust
- Rollinson, A. N., & Karmakar, M. K. (2015). On the reactivity of various biomass species with CO 2 using a standardized methodology for fixed-bed gasification. Chemical Engineering Science, 128, 82–91. http://dx.doi.org/10.1016/j.ces.2015.02.007 (Sum = 108.8%)
- Barrio, M., Gbel, B., Rimes, H., Henriksen, U., Hustad, J. E., & Srensen, L. H. (n.d.). Steam Gasification of Wood Char and the Effect of Hydrogen Inhibition on the Chemical Kinetics. Progress in Thermochemical Biomass Conversion, 32–46. http://dx.doi.org/10.1002/9780470694954.ch2 (Sum = 99.6%)
- Rukshan Jayathilake and Souman Rudra. (2017). Numerical and Experimental Investigation of Equivalence Ratio (ER) and Feedstock Particle Size on Birchwood Gasification. Energies, 10(8), 1232. http://dx.doi.org/10.3390/en10081232 (Sum = 99.5%)
- S. Sarker, H. K. Nielsen. Preliminary fixed-bed downdraft gasification of birch woodchips. Int. J. Environ. Sci. Technol. (2015) 12:2119–2126. http://dx.doi.org/10.1007/s13762-014-0618-8 (Sum = 99.5%)
- Zhou, C., Rosén, C., & Engvall, K. (2016). Biomass oxygen/steam gasification in a pressurized bubbling fluidized bed: Agglomeration behavior. Applied Energy, 172, 230–250. http://dx.doi.org/10.1016/j.apenergy.2016.03.106 (Sum = 99.6%)
To be corrected in the fifth column: it is "Ah", it should be "Ash".
Done, thank you.
Fig. 3
Please refer to the value in the area 0.8 <Φ> 1 (in the context of the process).
We added a sentence about the composition of detonation products of the fuel-lean mixture with Phi=0.95:
“At Φ ≤ 1, the detonation products were mainly composed of H2O and CO2 with nearly zero fractions of H2 and CO and nonzero fraction of O2, which attained 15%vol.d.b. for the 100-dm3 reactor at Φ = 0.95.”
Chapter 3.2.1 Liquid waste and Chapter 3.2.2 Solid waste
How were the temperature zones in the reactors distributed: 100 dm3 and 40 dm3?
Based on the snapshots like those shown in the new Fig. 2, we expect that the maximum gas temperatures in the 40-liter and 100-liter reactors were nearly the same, but the mean gas temperature in the 40-liter reactor was considerably higher than in the 100-liter reactor. We addressed this issue in the additional text discussed earlier in this response to the reviewer:
“The obtained temperature difference comes from the larger ratio of the PD tube volume to reactor volume: in the 40-dm3 reactor, hot products of the new cycle were diluted in a smaller volume of expanded products of the previous cycle, as compared to the 100-dm3 reactor. Due to very high velocities of incoming detonation products (over 1 km/s), the shape of reactor (spherical or cylindrical) does not play an important role. Moreover, according to [31], the wall temperature and therefore the wall surface area do not play a significant role in the time history of mean temperature.”
Chapter 3.2.1 Liquid waste
Quote: „……no mineral residues were detected.”
Comment: Please explain this because table 2 shows the value of ash as 0.7%.
It is difficult to explain indeed because we did not detect the ashes systematically, probably, due to a relatively large volume of the manifolds (tubing, cyclone, cooler, etc.) attached downstream of the reactor: the microparticles of ash could be accumulated at the internal surfaces. We added a sentence about it:
“In the liquid waste gasification experiments, no condensed-phase by-products like tar and char, as well as no mineral residues were systematically detected. Probably, this was caused by a relatively large surface area of manifolds attached downstream from the reactor which could accumulate condensed-phase microparticles.”
Chapter 3.2.2 Solid waste
Quote: „……no mineral residues were detected.”
Comment: Please explain this as table 2 shows the value of ash as 0.5%.
As for the lack of mineral residues for solid wastes, we have added a sentence to the subsection 3.2.2:
“In the solid waste gasification experiments, no condensed-phase by-products like tar and char, as well as no mineral residues were systematically detected. Similar to the liquid waste, this was probably caused by a relatively large surface area of manifolds attached downstream from the reactor which could accumulate condensed-phase microparticles.”
Chapter 4. Discussion
1. Quote: „……mineral residues were not detected in the experiments on liquid/solid waste gasification……..Firstly, the feedstocks contained nearly no ash.”
Comment: It's not true. The ash amounts in Table 2 show that after gasification of e.g. 2-3 kg of solid waste (chapter 3.2 Liquid / solid waste gasification), 10-15 g of ash remains, which is a traceable amount. The same applies to liguid waste containing 0.7% ash.
We have reformulated the corresponding sentence as
“Firstly, the feedstocks contained small amounts of ash, which could be accumulated in the form of condensed-phase microparticles at internal surfaces of manifolds attached downstream from the reactor.”
2. Quote: „In the absence of free oxygen, the gasification processes of liquid/solid wastes are known to include heterogeneous (endothermic) reactions:
C + H2O = CO + H2
C + CO2 = 2CO
C + 2H2 = CH4
and gas-phase reactions:
CO + H2O = CO2 + H2
CnHm + nH2O = nCO + (n+0.5m)H2
CnHm + nCO2 = 2nCO + 0.5mH2”
Comment: There are errors. The reaction C + 2H2 = CH4 is an exothermic reaction: –74,9 kJ/mol. Also the reaction CO + H2O = CO2 + H2 is an exothermic reaction: –41,2 kJ/mol.
Firstly, we moved the list of reactions to the Introduction section. Secondly, we have reformulated this sentence as:
“In the absence of free oxygen, the gasification processes of liquid/solid wastes are known to include endothermic and exothermic heterogeneous reactions [2-4]:”
It is unclear why the authors did not report exothermic reactions, which also occur e.g:
2C + 2H2O = CH4 + CO2 -8,79kJ/mol
CO + 3H2 = CH4+H2O -205kJ/mol
CO2 + 4H2 = CH4+ 2H2O –167,3 kJ/mol
2CO + 2H2 = CH4+ CO2 –247 kJ/mol
Our intention was to show the most important routes of gasification reactions. Nevertheless, following the reviewer’s comment we have extended the list of reactions. Now the corresponding text is:
The gasification processes of liquid/solid wastes in the H2O/CO2 environment, i. e., in the absence of free oxygen, are known to include endothermic and exothermic heterogeneous reactions [2-4]:
C + H2O = CO + H2
C + CO2 = 2CO (1)
C + 2H2 = CH4
2C + 2H2O = CH4 + CO2
and gas-phase reactions:
CO + H2O = CO2 + H2
CO + 3H2 = CH4 + H2O
CO2 + 4H2 = CH4+ 2H2O
2CO + 2H2 = CH4+ CO2 (2)
CnHm + nH2O = nCO + (n+0.5m)H2
CnHm + nCO2 = 2nCO + 0.5mH2
The rates of these reactions are highly sensitive to the gasification temperature, while the degree of their completion depends on the gas and condensed-phase residence times in the flow reactors.

Round 2
Reviewer 1 Report
Authors succesfully responsed my previous enquiries. I also think the newly added data provides enough enhancement of the work to be published in its current form